# Transcriptome Analysis Reveals Novel Genes Potentially Involved in Tuberization in Potato

**DOI:** 10.3390/plants13060795

**Published:** 2024-03-11

**Authors:** Meihua Zhang, Hongju Jian, Lina Shang, Ke Wang, Shiqi Wen, Zihan Li, Rongrong Liu, Lijun Jia, Zhenlin Huang, Dianqiu Lyu

**Affiliations:** 1Integrative Science Center of Germplasm Creation in Western China (CHONGQING) Science City, Southwest University, Chongqing 401329, China; 18323363829@163.com (M.Z.); hjjian518@swu.edu.cn (H.J.); cxldshanglina@163.com (L.S.); carnival@email.swu.edu.cn (S.W.); 15953919095@163.com (Z.L.); liurongrong98@outlook.com (R.L.); jialijun1224@162.com (L.J.); 2College of Agronomy and Biotechnology, Southwest University, Chongqing 400715, China; 3Engineering Research Center of South Upland Agriculture, Ministry of Education, Chongqing 400715, China; 4Chongqing Key Laboratory of Biology and Genetic Breeding for Tuber and Root Crops, Southwest University, Chongqing 400715, China; 5Chongqing Agricultural Technical Extension Station, Chongqing 401121, China; hzlcqzbz@163.com

**Keywords:** potato, transcriptome sequencing, tuberization, WGCNA, network

## Abstract

The formation and development of tubers, the primary edible and economic organ of potatoes, directly affect their yield and quality. The regulatory network and mechanism of tuberization have been preliminarily revealed in recent years, but plenty of relevant genes remain to be discovered. A few candidate genes were provided due to the simplicity of sampling and result analysis of previous transcriptomes related to tuberization. We sequenced and thoroughly analyzed the transcriptomes of thirteen tissues from potato plants at the tuber proliferation phase to provide more reference information and gene resources. Among them, eight tissues were stolons and tubers at different developmental stages, which we focused on. Five critical periods of tuberization were selected to perform an analysis of differentially expressed genes (DEGs), according to the results of the tissue correlation. Compared with the unswollen stolons (Sto), 2751, 4897, 6635, and 9700 DEGs were detected in the slightly swollen stolons (Sto1), swollen stolons (Sto2), tubers of proliferation stage 1 (Tu1), and tubers of proliferation stage 4 (Tu4). A total of 854 transcription factors and 164 hormone pathway genes were identified in the DEGs. Furthermore, three co-expression networks associated with Sto–Sto1, Sto2–Tu1, and tubers of proliferation stages two to five (Tu2–Tu5) were built using the weighted gene co-expression network analysis (WGCNA). Thirty hub genes (HGs) and 30 hub transcription factors (HTFs) were screened and focalized in these networks. We found that five HGs were reported to regulate tuberization, and most of the remaining HGs and HTFs co-expressed with them. The orthologs of these HGs and HTFs were reported to regulate processes (e.g., flowering, cell division, hormone synthesis, metabolism and signal transduction, sucrose transport, and starch synthesis) that were also required for tuberization. Such results further support their potential to control tuberization. Our study provides insights and countless candidate genes of the regulatory network of tuberization, laying the foundation for further elucidating the genetic basis of tuber development.

## 1. Introduction

Potato (*Solanum tuberosum* L.), a crucial food and vegetable crop, has strong adaptability and high nutritional value. It is widely planted worldwide, improving global food security and nutritional levels. The formation and development of tubers directly affect potato yield and economic benefits. Research on the regulation mechanism of tuberization has attracted significant attention. Tuber is a modified subterraneous stem formed by the swelling of the subapical region of the stolon, which has the dual functions of storage organ and vegetative propagation system. Tuberization is a continuous and complex process, mainly including the formation and elongation of stolons, the induction of tuber initiation, and tuber proliferation. Under the conditions of tuber induction, the growth of stolon apical meristem cells becomes determinate, and cell division ceases after a few rounds, resulting in the arrest of stolon elongation. However, cell division and expansion occur in specific cell layers between the cortex and pith of the subapical region of the stolon [1]. The orientation of cell division changes from transverse to longitudinal, leading to the radial expansion of the subapical region of the stolon. As a result, the stolon subapical region acts as a strong sink. The sucrose unloading mechanism of the stolon shifts from apoplastic to symplastic, accumulating plentiful starch and storage proteins until the tubers fully mature [2].

The stolon-to-tuber transition is a critical developmental phase influenced by many factors, such as light, temperature, moisture, sucrose, nitrogen, and phytohormone, in addition to its genetics. Studies have shown that short days, cool temperatures, high light intensity, high sucrose content, and low nitrogen application rates can promote tuberization. In contrast, long days, elevated temperatures, low light intensity, and high nitrogen application rates can delay potato tuber formation [3,4]. Phytohormones are essential regulatory factors of tuberization. Previous reports indicated that gibberellin (GA) and strigolactone (SL) inhibit tuberization [5,6]. In contrast, cytokinin (CK), auxin (IAA), and abscisic acid (ABA) are beneficial to tuber formation [7,8,9]. Additionally, the interaction between distinct factors, such as sugars and hormones, significantly influences tuber formation [10].

The diverse influencing factors make the regulatory network of tuberization intricate. Recently, the regulation mechanisms of tuberization by photoperiods and hormones have been preliminarily elucidated [1,11]. *StPHYB* (*PHYTOCHROME B*), *StSP6A* (*SELF-PRUNING 6A*), *StCO* (*CONSTANS*), *StCDF1* (*CYCLING DOF FACTOR 1*), and *StBEL5* (*BEL1-LIKE TRANSCRIPTION FACTOR 5*) are the five most essential signal molecules that regulate tuberization by photoperiods. Among these, *StSP6A*, *StCDF1*, and *StBEL5* are positive regulators of tuber formation, while *StPHYB* and *StCO* negatively regulate tuberization [1,12]. GA is a potent repressor of tuber initiation. *StGA2ox1* (*GIBBERELLIN OXIDASE 2OX1*), as a critical metabolic gene of GA, was upregulated during the initial stages of tuber development before visible swelling and predominantly expressed in the subapical region of the stolon and growing tuber. The 35S-overexpression transformants exhibit a dwarf phenotype, reduced stolon growth, and earlier in vitro tuberization [13]. However, overexpression of its synthetic gene *StGA20ox1* (*GIBBERELLIN OXIDASE 20OX1*) caused opposite changes. StPOTH1 (POTATO HOMEOBOX 1) forms a heterodimer with StBEL5, which binds to tandem TTGAC motifs in *StGA20ox1* promoters and has been shown to reduce the GA levels required for tuber formation [14].

Moreover, the role of auxin in tuberization was demonstrated by the early potato formation phenotype in vitro after auxin application [15]. Potato plants expressing the cytokinin biosynthesis gene *StIPT* (*ISOPENTENYL TRANSFERASE 1*) yielded more tubers but with reduced tuber weight. A microarray-based expression study showed that many auxin-related genes were differentially expressed during early tuber developmental stages, such as auxin transport (PIN gene family), auxin response factors (ARFs), and Aux/IAA genes [13]. Auxin production and directional transport systems in stolons also act synergically with SL to control secondary stolon growth, similar to controlling aboveground branches [6]. Stolons develop from underground axillary buds, and their number directly affects the number of harvested tubers. The TCP/TB1 transcription factor *StBRC1B* (*BRANCHED 1B*) inhibited aerial tuber formation by regulating axillary bud activity [16]. In the initiating tubers, sucrose is transported from the leaves, and abundant starch and stored protein accumulate, prompting the tuber to expand and mature rapidly. *StSUSY1* (S*UCROSE SYNTHASE 1*), *StSUT1* (*SUCROSE TRANSPORTER 1*), and *StAGPase* (*ADP-GLUCOSE PYROPHOSPHORYLASE*) have been shown to regulate tuber development by affecting the balance of source–sink [11,17]. We have recently gained a significant understanding of the morphological process and many signaling regulatory molecules involved in tuber formation. Nonetheless, many genes still need to be discovered to elucidate the regulatory pathways of tuberization further [1].

As an autotetraploid species that depends on tuber reproduction, the potato has serious self-incompatibility and inbreeding depression, which makes it challenging to study gene function using forward genetics. Transcriptome sequencing is one of the essential and effective strategies for screening regulatory molecules corresponding to certain traits, and it is widely used in expression profile analysis related to potato biotic and abiotic stresses, nitrogen stress, hormones, and development [18,19,20,21,22]. In addition, the transcriptome is also used in the studies of tuberous root development. Although the development process of tuberous root is different from that of tuber, it shares some metabolic processes and regulatory molecules [2]. Transcriptome studies of tuberous root at three developmental stages found that some specific signal transduction pathways and metabolic processes like plant hormone signal transduction, Ca^2+^ signa, starch and sucrose metabolism, and cell cycle and cell wall metabolism are related to tuberous root development in sweet potato [23]. However, there are few transcriptome studies on the process of tuber formation and development. Transcriptome analysis of the grafted plants of the interference line and wild type showed that StPHYF may participate in regulating the circadian clock, source–sink relationship to modulate tuberization [24]. The transcriptome expression profiles of three different tuberization-specific clones under different photoperiods found that genes in the pathway of circadian rhythm, signal transduction, and development play a role in the photoperiod regulation of tubers and indicated that tubers may be primarily controlled by homologous genes that regulate flowering time in other plants [25]. Moreover, transcriptome analysis of tubers at three developmental stages found that DEGs were mainly enriched in energy and carbohydrate metabolism pathways [26]. However, the tissues and stages of sampling in these studies were elementary, and the analysis of these transcriptomics was limited to the number of differentially expressed genes and enriched pathways. Atlantic is a high-quality potato cultivar that is widely planted in the United States, Canada, Europe, and Asia. It has the advantages of high yield, shallow eye, and storage resistance. Therefore, intensive and detailed sampling and transcriptome analysis of the Atlantic tissues were conducted in our study. The genes, transcription factors, and hormone-related genes specifically and differentially expressed in stolons and tubers were collected and analyzed. The co-expression networks of different tuberization stages were constructed by WGCAN and DEG analyses. Furthermore, the HGs and HTFs of these networks were screened, which may play crucial regulatory roles in the corresponding stages of tuber formation and development. In summary, our research provided reliable and valuable bioinformatics resources for a profound understanding of tuberization and precise control of crop improvement.

## 2. Results

### 2.1. Global Analysis of Transcriptome

To dissect the underlying molecular regulatory networks of tuberization, thirteen tissues of potato plants were taken for transcriptome sequencing using the Illumina platform. After quality control of raw reads, 26 samples obtained 39.46–56.07 million clean reads. Among them, 0.53–14.41 million clean reads that matched the ribosome library were removed from transcriptome analysis. Of the final 35.47–55.13 mRNA clean reads, 86.28–90.99% aligned to the reference genome. Unique mapped reads accounted for 78.95–88.0%. The results of the base quality analysis showed that the Q30 of each sample was higher than 93.91% (Appendix A). The expression data of 43,129 genes were obtained in each sample. Genes with FPKM < l were considered unexpressed, and 39.1%–47.8% of the genes were expressed in 13 tissues. The proportion of genes with 20 ≤ FPKM in each tissue was increased during stolon swelling (Sto1–Sto2) but decreased gradually with the expansion of tubers (Tu1–Tu5) (Figure 1B), which suggested that many medium- and high-expression genes were changed during the development of stolons and tubers.

Plant genes exhibit spatiotemporal expression differences to support different tissues that perform different functions. The cluster dendrogram showed that except for Rt and Sto–Sto2, other organs were clustered into different classes (Figure 1C). The development of stolon and tuber was a continuous process. However, according to the results of tissue correlation analysis, the gene expression profiles of stolons and tubers at different stages were widely divergent (0.21 ≤ R ≤ 0.79). The strong correlation of Tu1 and Tu2–Tu3 (0.94 ≤ R ≤ 0.98), Tu4, and Tu5 (R = 0.97) were observed, while they were poorly correlated between other stages. Interestingly, the gene expression of Sto–Sto2 exhibited numerous similarities with Rt and St (0.65 ≤ R ≤ 0.85), especially with Sto. Sto1–Sto2 presented modest similarity to SA (0.63 ≤ R ≤ 0.65) (Figure 1C,D). Meanwhile, the high consistency between the sample replicates was shown using a correlation analysis (Figure 1D).

### 2.2. Co-Expression Networks Associated with Thirteen Tissues

WGCNA is an important means to quickly discover hub genes highly associated with traits from multiple transcriptome data, which is widely used to screen genes related to specific traits, tissues, or developmental stages [27]. This study selected the gene sets with FPKM ≥ 1 in 13 tissues for WGCNA analysis, and 11 co-expression modules were obtained (Figure 2A). Among them, MEturquoise, MEdarkolivegreen, MEcoral1, MEblue, MEdarkgreen, MEdarkgrey, MElightsteelblue, andMEantiquewhite2 were associated with Le, Rt, St, Bud, SA, Sto–Sto1, Sto2–Tu1, and Tu2–Tu5, respectively, and the number of genes in each module ranged from 341 to 1942 (Figure 2A; Appendix A).

Furthermore, GO enrichment analysis was performed on the genes of these modules (Figure 2B; Appendix A). In most cases, genes in different modules were enriched in different pathways, and only a small number of pathways were enriched using two or more modules. The enriched pathways were closely related to the function of the tissues associated with the modules. Leaves are the site of photosynthesis and energy conversion, and the genes of the MEturquoise module were mainly enriched in photosynthesis (GO:0015979), starch metabolic processes (GO:0005982), and carbon fixation (GO: 0015977). Root, an organ responsible for absorbing water, inorganic salts, and nutrients. Genes, in its association module, were mainly enriched in response to nitrogen compound (GO:1901698), phosphorus metabolic process (GO:0006793), cell communication (GO:0007154), and fluid transport (GO:0042044). The cellular lipid metabolic process (GO:0044255), intracellular transport (GO:0046907), and protein transport (GO:0015031) were enriched by the MEcoral1 module, which was consistent with the function of the St in transportation. Genes in MEdarkgreen were mainly enriched in cell division (GO:0051301), meristem development (GO:0048507), floral organ morphogenesis (GO:0048444), shoot system development (GO:0048367), and auxin transport (GO:0060918), which supported the characteristic of SA as the origin site of flower primordia and bud primordia. Sto and Sto1 are tissues at the initiation of tuber induction. Their cell division orientation changes from transverse to longitudinal [2]. Correspondingly, many genes in MEdarkgrey were enriched in the beta-glucan metabolic process (GO:0051273), cellulose metabolic process (GO:0030243), and radial pattern formation (GO:0009956). Sto2 and Tu1 were tissues that were swelling and transforming into tubers, and extensive division and differentiation were proceeding in their cells to support further enlargement. Consistently, genes in the MElightsteelblue were enriched in cell wall organization or biogenesis (GO:0071554), tissue development (GO:0009888), regulation of meristem development (GO:0048509), and nitrogen utilization (GO:0019740). Genes in the MEantiquewhite2 were mainly enriched by the negative regulation of hydrolase activity (GO:0051346), glycogen metabolic process (GO:0005977), energy reserve metabolic process (GO:0006112), and starch metabolic process (GO:0005982), which conformed with that Tu2–Tu5, as gradually expanding and maturing tubers, were strong sinks for storing starch and protein. Interestingly, MEdarkolivegreen and MEdarkgrey were both enriched in anion transport (GO:0006820), and MEdarkolivegreen and Melightsteelblue were both enriched in the regulation of meristem development (GO:0048509). This implied that the tissues corresponding to these modules may have some similar functions.

### 2.3. DEGs and DETFs of Stolons and Tubers in Different Developmental Stages

Based on the tissue correlation analysis, a strong correlation between Tu1 and Tu2–Tu3 (0.94 ≤ R ≤ 0.98), Tu4, and Tu5 (R = 0.97) was observed (Figure 1C). Therefore, tubers of five stages were divided into two categories, and Tu1 and Tu4 were selected for subsequent analysis. Therefore, DEG analysis was performed for the five critical periods of Sto, Sto1, Sto2, Tu1, and Tu4. Sto vs. Sto1, Sto vs. Sto2, Sto vs. Tu1, and Sto vs. Tu4 contained 4271, 4897, 6635, and 9700 DEGs, respectively. There were about twice as many downregulated DEGs as upregulated DEGs in Sto vs. Tu4 (Figure 3A). This may be because Tu4, as tubers in the later developmental stage, mainly stored starch, while genes that play functions such as cell division and differentiation in the early stages were no longer needed. The number of DEGs that differed among the comparison combinations was counted, and a Venn plot was drawn (Figure 3B). The results showed that 12.05% of DEGs were shared by four comparison combinations. Moreover, 5.4%, 2.69%, 6.5%, and 28.57% of DEGs were unique to Sto vs. Sto1, Sto vs. Sto2, Sto vs. Tu1, and Sto vs. Tu4, respectively.

Transcription factors play critical roles in plant development. A total of 854 DETFs were selected in the stolons and tubers at different developmental stages. Sto vs. Sto1, Sto vs. Sto2, Sto vs. Tu1, and Sto vs. Tu4 contained 328, 300, 434, and 611 DETFs, respectively. A total of 78.81% (673/854) DETFs were present in two or more comparison combinations, and the dynamic change in these DETFs was hypothesized to be critical for tuberization (Figure 3C). DETFs belonged to 49 families, of which the ERF, bHLH, MYB, WRKY, and C2H2 families accounted for the largest proportion (40.94%) (Figure 3D). GO enrichment analysis showed that these DETFs were mainly enriched in tissue development (GO:0009888), meristem development (GO:0048507), cell division (GO:0051301), cell differentiation (GO:0030154), shoot system development (GO:0048367), floral organ development (GO:0048437), response to hormone (GO:0009725), and gibberellin-mediated signaling pathway (GO:0010476) (Figure 3E). These pathways were accompanied by stolon-to-tuber or directly regulated the process of tuberization. In conclusion, these results indicate that the DETFs play an important role in tuberization.

### 2.4. DEGs of Hormone Regulation Pathway

Phytohormones are important factors affecting the initiation and development of stolons and tubers. A total of 164 hormone pathway genes were obtained in DEG sets. As an inhibitor of tuber formation, GA decreased rapidly when the stolons transitioned to tubers [28]. At different stages of tuberization, 12 DEGs were identified to be involved in GA synthesis, metabolism, and signal transduction. All GA metabolism genes were rapidly upregulated at stolon swelling (Sto1–Sto2), and *StGA2ox1* (*PGSC0003DMG400021095*) remained upregulated at tuber enlargement. In contrast, GA synthesis genes were consistently downregulated or upregulated during tuber proliferation (Figure 4A). This expression pattern resulted in low GA levels at tuber formation, consistent with previous studies. As a promoter of tuberization, the IAA content was high in the stolon tip at the tuber initiation stage, but it decreased during the later stages of tuber development [6]. Thirty DEGs were identified to be involved in IAA metabolism and signal transduction, and 90% were involved in signal transduction pathways. Among them, 51.85% (14/27) were upregulated during tuberization, At the same time, the rest were downregulated at the later stage of tuber proliferation (Figure 4A), which was consistent with the accumulation level of IAA (Figure 4B). At different developmental stages, 45 DEGs were found to be involved in ABA synthesis, metabolism, and signal transduction. The four IAA synthetic genes were all upregulated. Among the 40 signal transduction pathway genes, 60% (24/40) were upregulated during tuberization, 27.5% (11/40) were downregulated in the late stage of tuber proliferation, and the remaining five genes were always downregulated (Figure 4C). Its role as a positive tuber formation and development regulator confirmed this result. Moreover, many DEGs were involved in the synthesis and signal transduction pathways of tuber-inducible hormones BR, CK, ETH, and JA, which supported their role in the regulation of tuberization (Figure 4D–G). At the same time, in the signal transduction pathway of salicylic acid (SA), a phytohormone that has not been reported to control the tuberization process, 15 genes were differentially expressed in tuber formation and expansion, suggesting its possibility to regulate tuberization (Figure 4H).

### 2.5. DEGs of Reported Tuberization Regulatory Network

Previous studies have preliminarily revealed the tuberization regulatory network influenced by photoperiods, hormones, and sucrose. We extracted 41 genes from this network for analysis (Appendix A). The results showed that both positively and negatively regulated genes had tissue expression specificity (Figure 5). Among these genes, 56.1% (23/41) were specifically expressed in stolons or tubers, and 87.8% (36/41) were differentially expressed in different developmental stages. Therefore, we suggested that the majority of genes controlling tuber development were specifically or differentially expressed in stolons and tubers. *StPatatin*, *StBEL5,* and *StGA2ox1* were highly expressed in Tu1–Tu5, as presented in the set of upregulated DEGs in Sto vs. Tu1. *StSP6A*, *StIAA3* (*INDOLEACETIC ACID-INDUCED PROTEIN 3*), *StARP* (*AUXIN RESPONSE PROTEIN*), and *StSUSY1* (*SUCROSE SYNTHASE 1*) were highly expressed in Sto–Sto2. Among these, the expressions of *StSP6A* gradually decreased in Sto1–Sto2 compared with Sto, while the expressions of *StIAA3*, *StARP*, and *StSUSY1* were upregulated in Sto1–Sto2 (Figure 5 and Figure 3A). Mover, most repressors maintain low expression levels in stolons or tubers, such as *StGA20ox1*, *StCO2,* and *StPTH15* (Figure 5). The expression patterns of the genes above were consistent with their functions and previous research. However, a few genes that did not match the expectations, such as the expression of the negative regulatory genes *StBRC1B* and *StBRC2B* in potatoes were upregulated in Sto1 and Sto2. The possible reason was that these two genes negatively regulated the number of tubers by inhibiting the axillary bud activity and branching formation of stolons [16], and exerted inhibitory effects only before the swelling of stolons. In contrast, the upregulated expression after swelling could inhibit the formation of branching and promote the further expansion of stolons. In summary, most reported genes were differentially expressed during the tuberization process, and their expression patterns were consistent with previous studies. These results indicate that our transcriptome data have a significant reference value for exploiting candidate genes of tuberization.

### 2.6. Co-Expression Networks’ Construction of Stolons and Tubers Associative Modules

Through WGCNA analysis, MEdarkgrey, Melightsteelblue, and MEantiquewhite2 were screened and associated with Sto–Sto1, Sto2–Tu1, and Tu2–Tu5, respectively (Figure 2A). The high-ranking ten genes and TFs with KME values in the module were recognized as HGs and HTFs. The co-expression networks were constructed using the HGs and HTFs and the DEGs in the modules. There were 730 genes and 39 TFs in the MEdarkgrey network, which contained reported tuberization genes *StSUT1*, *StCesA2* (*CELLULOSE SYNTHASE*), *StBRC1B*, *StAGL8* (*POTM1-1*), *StSUSY1*, and *StPOTH1* (Appendix A). *StSUT1* was co-expressed with 48.23% of the genes in the network, all HGs, and 50% HTFs (Figure 6A; Appendix A). The MElightsteelblue network contained 313 genes and 29 TFs. The positive regulatory genes *StBRK* (*BRASSINOSTEROID KINASE*) was co-expressed with 8.33% of the genes in the network, including 90% HGs and 70% HTFs (Figure 7A; Appendix A). The MEantiquewhite2 network had 539 genes and 24 TFs. It contained the tuberization genes *StAGPase*, *StPatatin*, and *StPT3* (*PURINE TRANSPORTER 3*), which were co-expressed with 44.53% of the genes in the network, including all HGs and HTFs (Figure 8A; Appendix A). To verify the accuracy of sequencing data and networks, the expression patterns of HGs and HTFs in different tissues were detected by qRT-PCR. The results showed that the transcriptome data and networks partitioning were reliable (Figure 6B, Figure 7B and Figure 8B). Furthermore, in RH89-039-16 transcriptome data (http://spuddb.uga.edu/pgsc_download.shtml, accessed on 2 March 2024), most of these genes were differentially expressed during tuber formation and development and were highly expressed in stolons or tubers (Appendix A). The high similarity of gene expression patterns in the two transcriptomes further confirmed the reliability of the results of this study.

Since the functions of most HGs and HTFs were unknown in potato, functional annotations of their Arabidopsis homologs were queried to predict their roles in tuberization (Appendix A). The result showed that these orthologs played crucial roles in regulating flowering, cell division and tissue differentiation, branching, hormone synthesis, metabolism and signal transduction, sucrose transport, carbon assimilation, starch synthesis and metabolism, and cellulose deposition. These physiological processes and pathways are also essential for developing stolons and tubers. In summary, 30 HGs and 30 HTFs were selected from the three co-expression networks of stolon or tuber association, most of which were co-expressed with the reported tuberization genes in the networks. The regulatory processes involved in their homologs were also required for tuberization. Therefore, these HGs and HTFs were considered to have great potential to regulate the tuberization process.

## 3. Discussion

Tuber, which develops from the subapical region of stolon, is a crucial asexually reproductive and edible organ of potato. Studies on the physiological and biochemical processes and molecular mechanisms of tuberization are profound for breeding excellent potato varieties. The tuberization regulatory network has been preliminarily revealed, but it still needs to be further enriched and improved [1,11]. Transcriptome sequencing technology based on a high-throughput sequencing platform is an efficient and rapid research method that can comprehensively obtain the transcript information of specific tissues or organs and has guiding significance for excavating the related regulatory factors of a particular tissue or treatment. Previous transcriptome studies have provided essential references for the tuberization mechanism, but these studies sampled simple stages and tissues [25,26]. In this study, we sampled and sequenced 13 tissues from potato plants at the tuber proliferation phase to obtain more detailed reference data, hoping to support the study of the tuberization mechanism.

It is well known that during the growth and development of all organisms, gene expression exhibits strict temporal and spatial specificity according to functional requirements [29,30]. The results of the tissue correlation analysis supported this viewpoint (Figure 1C,D). For example, as the site regarding photosynthesis and energy conversion, the highly expressed genes in leaves were mainly enriched in the biological processes of photosynthesis, starch metabolism, and carbon fixation. Roots function as mineral and water absorption and transport organs, in which highly expressed genes are mainly enriched in biological processes such as response to nitrogen compounds, phosphorus metabolic processes, and fluid transport. As an energy storage organ, most of the highly expressed genes in the tubers were enriched in the energy reserve and starch metabolic processes (Figure 2). Interestingly, we found that Sto–Sto1 showed high gene expression similarity with Rt (0.76 ≤ R ≤ 0.85) and St (0.72 ≤ R ≤ 0.73) (Figure 1D and Figure 2B). This similarity may be caused by the fact that Sto and Sto1 were not completely transformed into tubers as a unique stem, and the underground organ is the same as the root. The structures and functions of the four tissues were slightly resemblant. The genes in both MEdarkolivegreen (Rt) and MEdarkgrey (Sto–Sto1) were enriched in anion transport (GO:0006820), which also confirmed this hypothesis (Figure 2B). Moreover, the similarity between Sto1–Sto2 and SA (0.63 ≤ R ≤ 0.65) may be because Sto1 and Sto2 were the subapical tissues of stolons that were differentiating into tubers and had more active meristem cells like SA. MEdarkolivegreen (SA) and Melightsteelblue (Sto2–Tu1) were enriched in the regulation of meristem development (GO:0048509) supporting the results (Figure 1D and Figure 2B).

Tuberization includes the stages of stolon formation and elongation, induction of tuber formation, and tuber proliferation [31]. These stages were continuous, but there were numerous differences in morphology and gene expression (Figure 1A,C,D). Our analysis showed numerous DEGs and DETFs in different stages of stolons and tubers, which contained many reported tuberization regulatory genes and hormone pathway genes. The expression patterns of most reported genes were consistent with previous studies [1,11], which indicated that our transcriptome data were reliable for screening tuberization regulatory genes. Meanwhile, 854 DETFs belonging to 49 TF families were selected from the DEGs (Figure 3C–E). The transition from stolon to tuber is accompanied by cell division, differentiation, and elongation in the subapical region of the stolon [2,28]. Consistent with this, in our study, some DETFs were enriched in tissue development (GO:0009888), meristem development (GO:0048507), cell division (GO:0051301), and cell differentiation (GO:0030154) pathways (Figure 3E). The development of underground stolons is similar to that of the aboveground axillary branches of potatoes [32]. DETFs enriched in shoot system development (GO:0048367) were hypothesized to regulate tuberization by controlling the growth of stolons (Figure 3E). Tuber initiation and development were controlled by flowering genes [25,33]. Some DETFs were enriched in floral organ development (GO:0048437), supporting this viewpoint. GA and CK are essential for tuber initiation [5,7]. Consistently, in our study, the gibberellin-mediated signaling pathway (GO:0010476) and cytokinin metabolism (GO:0009690) were enriched by DETFs (Figure 3E). By comparison, previous transcriptome analysis has shown that DEGs are enriched in energy metabolism, carbohydrate metabolism, hormone signal transduction, and flower development during tuber development [25,26], consistent with our findings. In addition, we also found that many genes that were specifically expressed or differentially expressed in stolons and tubers were enriched in cell division and differentiation, branching system development, cellulose metabolic process, radial pattern formation, and nitrogen utilization process.

WGCNA is a systems biology method used to describe patterns of gene association between different samples, and designed to look for gene modules that are expressed collaboratively and to explore associations between gene networks and phenotypes of interest, as well as hub genes in the networks. Genes in the same co-expression network regulate the same process or have similar functions under specific conditions [27]. This study constructed three co-expression networks associated with stolons and tubers using WGCNA. Each network contained reported tuberization genes, which co-expressed with most HGs and HTFs (Figure 6, Figure 7 and Figure 8). These results suggested that the genes in the networks have the potential to regulate tuberization. The sucrose transporter StSUT1 promotes tuberization by participating in sucrose loading [17], which was co-expressed with all HGs and 50% HTFs in the MEdarkgrey (Sto–Sto1) network (Figure 6). Among these hub factors, Arabidopsis homologs of *StPLT5* and *StMYB61* play essential roles in sugar and organic matter partitioning [34,35]. They may be drawn to tuber-inducing roles by regulating the source–sink balance. When the stolon is induced to form a tuber, abundant assimilates are transported to the subapical region of the stolon along with the transport of hormones, in which vascular tissue is essential. In Arabidopsis, *MAMYB* and *NLP* control vascular tissue development [36,37]. This suggests that they have more potential to affect potato formation by regulating hormone and nutrient transport. Flowering and tuberization share many regulatory molecules [25,33]. Arabidopsis homolog genes of *StTPL*, *StEIN3*, *StHY5*, *StBEE1*, and *StDAP4* control flower development and flowering time [38,39,40,41,42,43]. Stolon-to-tuber transition is accompanied by incalculable cell division events and new cell formation, and cell wall formation is a hallmark of new cell morphogenesis in plants. Homologous *StCESA9*, *StDEAR3*, and *StERF043* consistently function in cellulose deposition and secondary cell wall synthesis [44,45,46,47]. Thus, sucrose transporter, hormonal signaling, flowering regulatory signaling, and genes controlling cell division are the major regulatory molecules during the early stages of stolon swelling.

Hormones directly affect stolon growth and tuber formation. The gene StBRK was contained in MElightsteelblue (Sto2–Tu1) and co-expressed with 90% HGs and 70% HTFs in the network (Figure 7), which took part in the IAA, BR, and CK regulatory networks to promote tuber development [48]. By querying Arabidopsis homologous gene functions, we found that 11 members of the HGs and HTFs were referred to as the synthesis, metabolism, or signaling transduction of hormones, such as GA, JA, ETH, IAA, and ABA (Appendix A). *BHLH93* is a crucial transcription factor for promoting flowering under non-inductive SD conditions through the GA signaling pathway [49]. In addition, PYL4 and *NLP7* in MEdarkgrey (Sto–Sto1) took part in the signal transduction of IAA, ABA, and JA to regulate plant growth and development [50,51,52]. Based on this, we suggest that hormones play a most important regulatory role in the early stages of tuber formation.

After tuber formation, perimedullary cells divide, expand, and store substantial amounts of starch until tuber maturation [28]. In the MEantiquewhite2 (Tu2–Tu5) network, *StAGPase*, *StPatatin*, and *StPR3*, which controlled tuber expansion and maturity by affecting energy metabolism, starch, and protein accumulation [11,53], were co-expressed with all HGs and HTFs (Figure 8). Among these HGs, the homologs of *StSBE2.2*, *StAATP1*, *StPHS1*, *StPK*, and *StAGPase* played roles in carbon assimilation and starch anabolism [54,55,56,57], where *StAGPase* has been shown to control tuber development by regulating carbon assimilation [53]. Furthermore, homologs of *StCDC20.1*, *StSCL28*, *StIDD9*, *StLRP1*, and *StBHLH36* have been reported to control cell division and tissue differentiation [58,59,60,61,62]. Tuberization was affected by nitrogen levels [3,22]. *HAT14*, an *HTF* in MEantiquewhite2, was reported to regulate nitrogen metabolism and plant growth [63], which suggested that they may regulate tuber formation by affecting nitrogen balance. Studies have shown that developing aboveground branches and underground stolons is similar [6]. As hub molecules of the MEantiquewhite2 network, the homologs of *StLOS2* and *StLOB* restrained branch development [64,65]. Therefore, molecules controlling cell division and tissue differentiation, energy storage, and metabolism were critical during the middle and later stages of tuber enlargement. Collectively, the vast majority of HGs and HTFs were co-expressed with reported tuberization genes. Their homologs were revealed according to the processes, such as regulating flowering, cell division and tissue differentiation, hormone synthesis, metabolism and signal transduction, vascular tissue development, sucrose transport, carbon assimilation, starch synthesis and metabolism, cellulose deposition, and secondary cell wall formation, which were also required for tuberization. These all indicate that HGs and HTFs in the three modules were candidate genes worth further investigation for controlling tuber formation.

## 4. Materials and Methods

### 4.1. Materials and Growth Conditions

The tetraploid cultivar ‘Atlantic’ was used as the experimental material in this study. The single-stem nodes of the plantlets were cultivated in MS medium with 4% sucrose for two weeks, and then the seedlings were hardened for two days. Next, the seedlings were planted in the greenhouse at Southwest University (29°48′52″ N, 106°25′16″ E), Chongqing, China, for 60 days. Roots (Rt), stems (St), leaves (Le), stem apex (SA), axillary buds (Bud), stolons at three swelling stages (Sto, Sto1, and Sto2), and tubers at five proliferation stages (Tu1, Tu2, Tu3, Tu4, and Tu5) were taken, respectively (Figure 1). After washing the soil, the samples were immediately frozen in liquid nitrogen and stored at −80 °C for later experiments.

### 4.2. Total RNA Extraction and cDNA Library Construction

Total RNAs of all samples were extracted using an EZ-10 DNAaway RNA Mini-Preps Kit (Sangon Biotech, Shanghai, China), and the quality of total RNAs was evaluated using an Agilent 2100 Bioanalyzer (SA Pathology, Adelaide, SA, Australia). High-quality RNAs (RIN > 8.0) were used for the following cDNA synthesis and library construction. After total RNA extraction, rRNAs were removed using the NEBNext RNA library preparation kit (#E7530, New England Biolabs, Ipswich, MA, USA), and mRNAs were enriched to construct cDNA library. Eukaryotic mRNAs with polyA tail were enriched using magnetic beads with oligo(dT), and the mRNAs were interrupted by ultrasound. The first cDNA strand was synthesized in the M-MuLV reverse transcriptase system using fragmented mRNAs as a template and random oligonucleotides as a primer. RNaseH degraded the RNA strand, and the second cDNA strand was synthesized in the DNA polymerase I system using dNTPs as the raw material. The purified double-stranded cDNA underwent end repair, A-tailing, and sequencing adapter ligation, and approximately 200 bp of cDNA was screened with AMPure XP beads and PCR amplification. PCR products were purified again using AMPure XPbeads to obtain a library. The DNA 1000 assay kit (5067-1504, Agilent Technologies, Shanghai, China) was used to assess the quality of the library.

### 4.3. High-Throughput RNA-Seq and Data Analysis

RNA sequencing was completed using GENE DENOVO (Guangzhou, China) Illumina sequencing. The sequencing method was paired-end sequencing, with each read length measuring 150 bp. The fastp tool was used to obtain clean reads and maintain quality control [66]. Clean reads were obtained by filtering out reads containing an adapter, N ratio greater than 10%, all A-base, mass value Q ≤ 20, and bases accounting for more than 50% of the entire read. In the genebank DNA sequence database of NCBI, the ribosome database was generated by obtaining the sequence with rRNA keywords in the annotation by searching for the genus name of potato. Clean reads were compared to the potato ribosome database using the short reads alignment tool bowtie2 [67]. The ribosome-matched reads were removed without allowing mismatches, and the remaining unmapped reads were used for subsequent transcriptome analysis. Using HISAT2 software (v2.1.0), alignment analysis based on the S. tuberosum Group Phureja clone DM1-3 (DM) v4.03 reference genome was conducted [68,69]. According to the alignment results of HISAT2, Stringtie was used to reconstruct the transcript, and the expression levels of all genes in each sample were calculated [70]. DESeq2 software (v1.20.0) was used to analyze the read count data [71], which ran with R (v3.6.0) in Rstutio. The read count was normalized, and the *p*-value was calculated according to the model. Finally, the false discovery rate (FDR) value was obtained using multiple-hypotheses testing corrections. The gene expression levels were calculated using fragments per kilobase of exon per million fragments mapped (FPKM). Genes with a *p*-value < 0.05 and |log2FC| >1 were screened as DEGs.

### 4.4. Gene Functional Enrichment Analysis

DEGs were mapped to each term of the Gene Ontology (GO) database (v3.8.2, http://www.geneontology.org/, accessed on 1 March 2020). The number of genes in each term was calculated to obtain the list of genes with a specific GO function and the number of genes. The Kyoto Encyclopedia of Genes and Genomes (KEGG) pathway was used pathway-significant enrichment analysis. A hypergeometric test was applied to determine the GO entries and pathways of significant enrichment in genes compared with the whole genomic background [72].

### 4.5. Construction of Weighted Gene Co-Expression Networks

Genes with FPKM > 1 in each sample were used for WGCNA using the R (v3.6.0) package WGCNA (v1.42) [73]. The power value was set to 10, the similarity was set to 0.75. The softConnectivity function was used to calculate the connectivity degree (KME) of the genes, and KME ≥ 0.7 was used as the threshold to screen the module genes. The 10 high-ranking genes in the module were selected as the hub genes (HGs) of the module, and the top 10 transcription factors with a KME value were selected as the hub transcription factors (HTFs). The Arabidopsis homologs of these HGs and HTFs were obtained through the NCBI alignment query of protein sequence, and their known functions were searched. The weight values between the different genes in the module were calculated according to the topological overlap matrix. The weight ≥ 0.1 was used as the threshold to screen the co-expressed genes. And the co-expressed genes that were not inquired in the DGE set were filtered out. Gephi 0.9.2 software was used to visualize the network [74].

### 4.6. Real-Time Quantitative PCR

Twenty-seven genes from the co-expression module associated with stolons or tubers were screened for qRT-PCR to ensure precise and reproducible sequencing results. cDNAs from eight tissue samples (Le, Rt, St, Sto, Sto1, Sto2, Tu1, and Tu4) were used in this experiment. Each gene and sample were repeated in triplicate. Elongation factor-1alpha (Stef1α) was used for each sample as an endogenous control [75]. BlazeTaq™ SYBR^®^ Green qPCR Mix 2.0 (GeneCopoeia, Rockville, MD, USA) was used for RT-qPCR reactions on a CFX96 Real-Time System (BIO-RAD, Hercules, CA, USA). Each reaction contained 1 μL of 10 × diluted cDNA strand, 5 μL of SYBR Green qPCR Mix, 4.6 μL of ddH2O, and 0.2 μL of each primer, accumulating a final volume of 10 μL in total. The formula F = 2^–ΔΔCt^ was used to calculate the relative expression levels. Specific primers were designed using Primer3web version 4.1.0 [76] and are shown in Appendix A.

## 5. Conclusions

In this study, transcriptome analysis was performed in 13 tissues of potato plants at the tuber proliferation phase. Among them, eight tissues were stolons and tubers at different developmental stages, which were focused. Numerous DEGs were screened at different developmental stages of stolons and tubers, which contained many TFs and hormone pathway genes. Simultaneously, three co-expression networks associated with stolons and tubers were constructed using WGCNA. Through analysis and discussion, we suggested that the HGs, HTFs, and co-expressed genes in these networks may regulate processes such as sucrose transport, hormone synthesis, metabolism, signal transduction, cell division, tissue differentiation, sink–source balance, energy storage, metabolism, and stolon branching, or serve as tuberization signaling to control tuber formation and development. The research results provide valuable references for further revealing and enriching the regulation network of tuberization.

## Figures and Tables

**Figure 1 plants-13-00795-f001:**
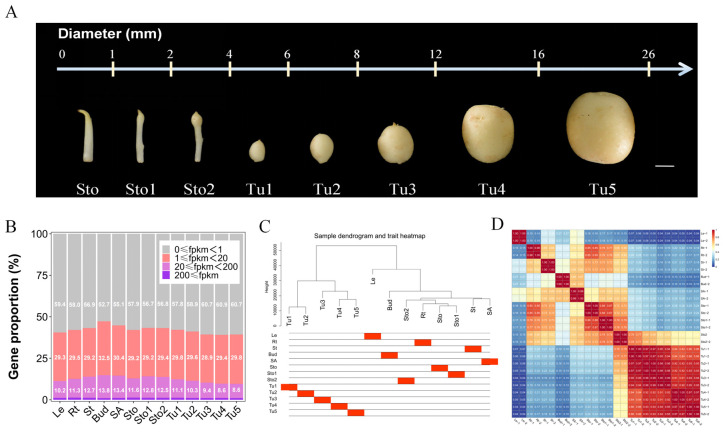
Global transcriptome analysis of 13 tissues from potato plants in the tuber proliferation phase. (**A**) Morphology of stolons and tubers at different development stages, bar = 5 mm. Sto, Sto1, and Sto2 were stolons at different swelling stages. Tu1, Tu2, Tu3, Tu4, and Tu5 were continuously expanding tubers. (**B**) Statistics and analysis of expression levels of all genes in each tissue. (**C**) The relationship between various tissues was exhibited in the sample dendrogram. (**D**) Pearson’s correlation coefficient analyses among 26 samples.

**Figure 2 plants-13-00795-f002:**
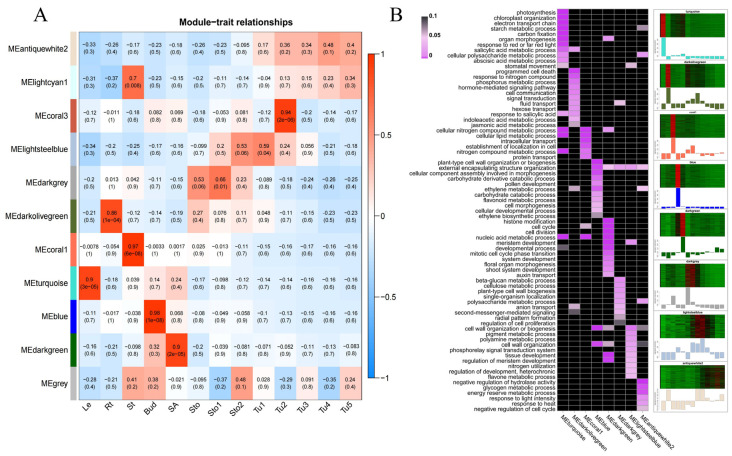
WGCNA of genes with FPKM ≥ 1 in 13 tissues. (**A**) Pearson’s correlation analysis of modules and tissues. (**B**) GO enrichment analysis of genes in modules associated with specific tissues. The heat map on the right shows the expression patterns of different module genes in various tissues, with red indicating high expression, black indicating moderate expression, and green indicating low expression. The color of the lower bar chart is used to distinguish different co-expression modules.

**Figure 3 plants-13-00795-f003:**
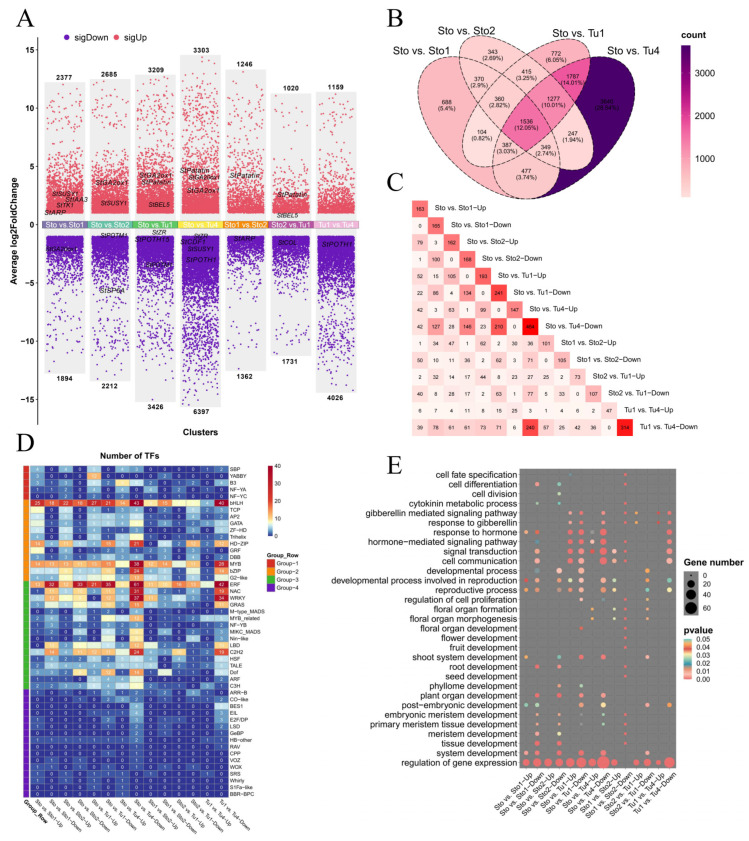
Comparative analysis of DEGs and DTFs. (**A**) The volcano plot demonstrates the upregulated and downregulated DEGs in each comparison combination. (**B**) Venn plot of DEGs between 5 developmental stages. (**C**) Heatmaps of upregulated and downregulated DETFs between 5 developmental stages. (**D**) Distribution of DETFs in each transcription factor family. (**E**) GO enrichment bubble plots of DETFs.

**Figure 4 plants-13-00795-f004:**
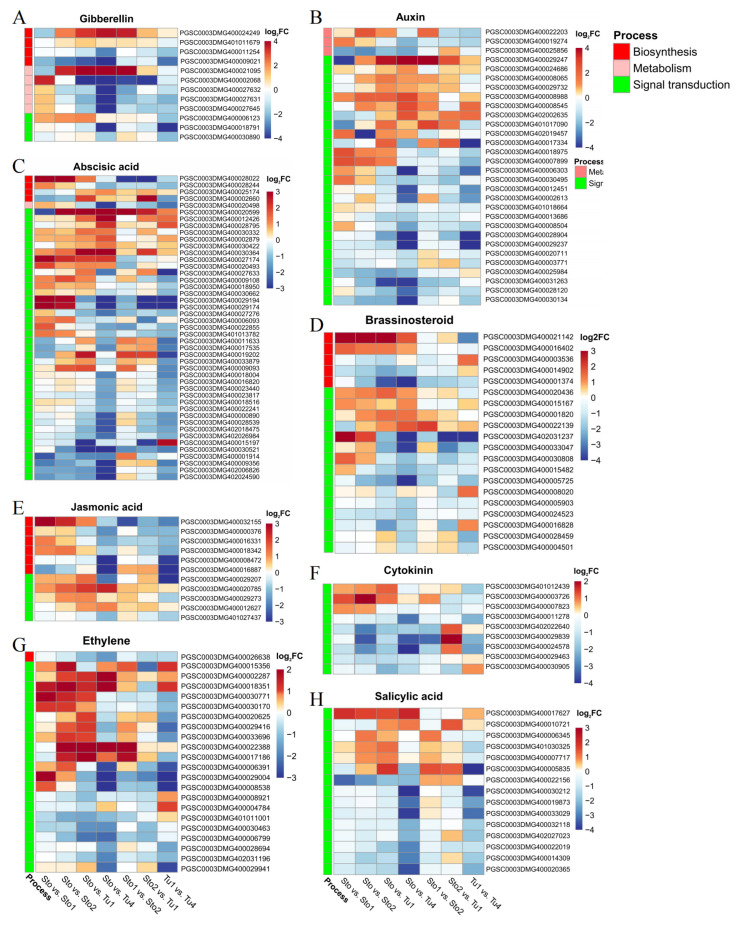
The DEGs of phytohormone pathways in stolons and tubers at different developmental stages. (**A**) Gibberellin. (**B**) Auxin. (**C**) Abscisic acid. (**D**) Brassinosteroid. (**E**) Jasmonic acid. (**F**) Cytokinin. (**G**) Ethylene. (**H**) Salicylic acid.

**Figure 5 plants-13-00795-f005:**
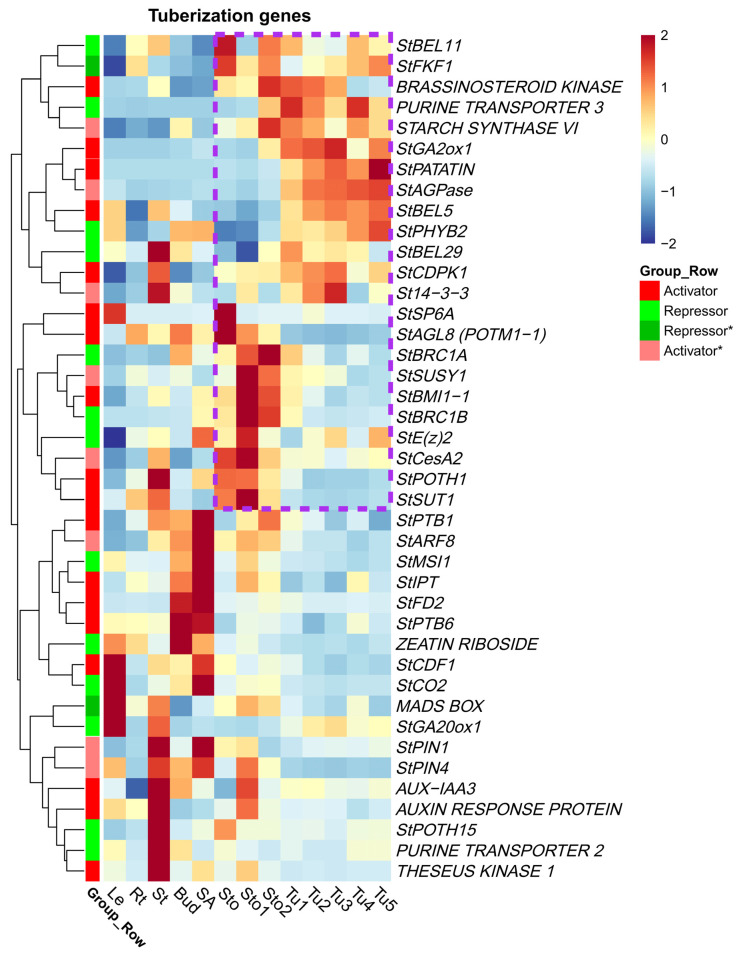
Expression patterns of reported tuberization genes in different tissues. Genes highly expressed in stolons or tubers were clustered through hierarchical clustering and highlighted by purple dashed lines. Repressors* and activators* represented the tuberization genes whose repressive or active function had not been verified by genetic means.

**Figure 6 plants-13-00795-f006:**
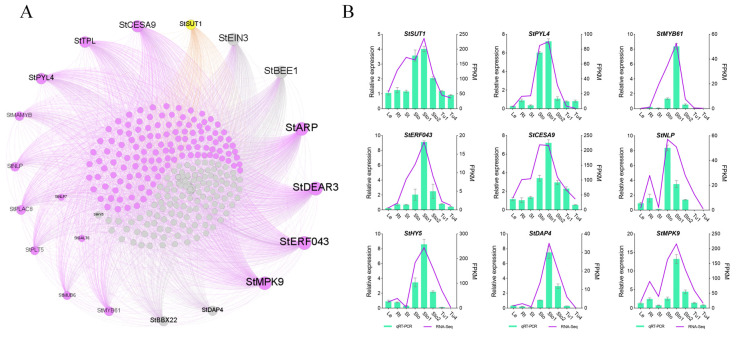
MEdarkgrey co-expression network associated with Sto–Sto1. (**A**) Co-expression network diagram. Yellow dots indicate reported tuberization genes. Purple and gray dots indicate genes co-expressed and not co-expressed with tuberization genes in the network, respectively. (**B**) qRT-PCR validation for genes in the MEdarkgrey co-expression network.

**Figure 7 plants-13-00795-f007:**
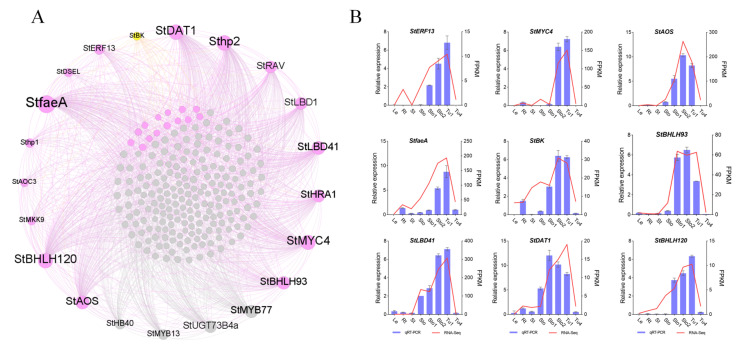
MElightsteelblue co-expression network associated with Sto2–Tu1. (**A**) Co-expression network diagram. Yellow dots indicate reported tuberization genes. Purple and gray dots indicate genes co-expressed and not co-expressed with tuberization genes in the network, respectively. (**B**) qRT-PCR validation for genes in the MElightsteelblue co-expression network.

**Figure 8 plants-13-00795-f008:**
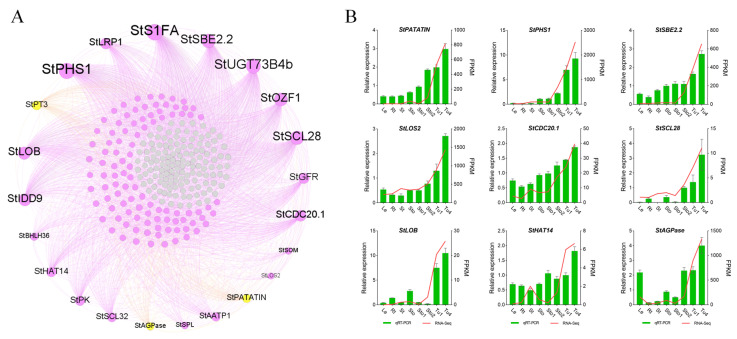
MEantiquewhite2 co-expression network associated with Tu2–Tu5. (**A**) Co-expression network diagram. Yellow dots indicate reported tuberization genes. Purple and gray dots indicate genes co-expressed and not co-expressed with tuberization genes in the network, respectively. (**B**) qRT-PCR validation for genes in the MEantiquewhite2 co-expression network.

## Data Availability

The datasets presented in this study can be found in online repositories. The names of the repository/repositories and accession number(s) can be found below: https://www.ncbi.nlm.nih.gov/bioproject/ PRJNA1035644 (accessed on 1 September 2023).

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
