# Peer review of "Transcriptome Analysis Reveals Novel Genes Potentially Involved in Tuberization in Potato"

_plants, 2024, doi:10.3390/plants13060795_

Round 1

Reviewer 1 Report

Comments and Suggestions for Authors

The results of this study provide information on the regulation of tuberization from stolon through transcriptome analysis at various stages of tuberization.

Major points:

1. However, despite providing a lot of information, it must be confirmed by comparing the results of expression analysis experiments such as qRT-PCR and Northern blot for DEGs, including 30 hub genes (HGs) and 30 hub transcription factors (HTFs), that show clear differences in the transcriptome analysis results.

2. References essential to explain the research results are missing. In addition, information about the researcher's results and comparative discussion should be added.

-BMC Genomics volume 23, Article number: 473 (2022)

-The Plant JournalVolume 113, Issue 2 p. 402-415

-Scientific Reports volume 10, Article number: 1152 (2020)

Minor points:

Line 23 ‘And a...’ -> ‘A’

Line 29 ‘And the..’ -> ‘The’

Author Response

We appreciate your critical review and encouraging appraisal of our manuscript entitled “Transcriptomic network analysis provides insights into the regulation of tuberization” (Manuscript ID: plants-2837595). We have revised our manuscript in accordance with the constructive criticism. The specific response to your comments are as follows:

  1. However, despite providing a lot of information, it must be confirmed by comparing the results of expression analysis experiments such as qRT-PCR and Northern blot for DEGs, including 30 hub genes (HGs) and 30 hub transcription factors (HTFs), that show clear differences in the transcriptome analysis results.

Response: Although we did not confirm the expression results of all HGs and HTFs, we used qRT-PCR to verify the expression differences of 27 members of them. The results were consistent with those of the transcriptome analysis, as mentioned in the manuscript results 2.6.

  1. References essential to explain the research results are missing. In addition, information about the researcher's results and comparative discussion should be added.

Response: Thank you for this valuable suggestion. We have supplemented the relevant contents in the revised manuscript in P15—P16.

Reviewer 2 Report

Comments and Suggestions for Authors

I have the following comments for this manuscript.

English language and syntax need improvement throughout the whole text. Certain sentences do not read well.

The title is very general and does not convey any scientific meaning. Please revise according to your objective and results.

There is no objective for this research. Authors, state clearly the objective(s) of this study, the reasons that you undertook the research and the innovations. I cannot find any innovative ideas on this manuscript.

Why did the authors use only the tetraploid cultivar ‘Atlantic’ as the experimental material in their study?

The gene expression analyses would have a greater strength if the authors have used at least two cultivars in order to validate and confirm the results of their research.

I find that the presentation of results is not well organized. There is a lot of data analysis and information that appear to have no relevance.

Comments on the Quality of English Language

Moderate editing of English language is required

Author Response

We extend our sincere gratitude for your valuable and constructive suggestions of the manuscript entitled “Transcriptomic network analysis provides insights into the regulation of tuberization” (Manuscript ID: plants-2837595). We have revised our manuscript in accordance with the constructive criticism. The specific reply to your comments are as follows:

1.  English language and syntax need improvement throughout the whole text. Certain sentences do not read well.

Response: Thank you for this valuable suggestion. We have made some improvements to the English language and grammar of the article. Meanwhile, we also asked the English polishing company to make modifications and submit the revised version in the next reply. 

2. The title is very general and does not convey any scientific meaning. Please revise according to your objective and results.

Response: According to your suggestion, we changed the title to ‘Transcriptome analysis reveals novel genes potentially involved in tuberization in potato’ in our new MS.

3. There is no objective for this research. Authors, state clearly the objective(s) of this study, the reasons that you undertook the research and the innovations. I cannot find any innovative ideas on this manuscript.

Response: 1) Reasons: firstly, there are currently few reports on the regulatory pathways involved in tuber development and the screening of candidate genes through transcriptome research. Secondly, the previous transcriptome sampling was conducted using plantlets or only three different developmental stages of tubers, and the sampled tissues were relatively rough. Furthermore, the analysis of these transcriptome results is limited to the number of differentially expressed genes and the analysis of enriched pathways. 2) Objectives and innovative ideas: our study conducted transcriptome analysis on 13 tissues of plants on the tuber proliferation phase. The main focus was on analyzing genes that are specifically expressed and differentially expressed in stolons or tubers. We not only conducted enrichment analysis on the DEGs, but also analyzed the differentially expressed transcription factors and hormone related genes. At the same time, we constructed a co-expression networks for three stages of tuber formation through WGCAN and DEG analysis, and selected HGs and HTFs from each network, which may play important roles in tuberization. In summary, the main objective of this study was to construct co-expression networks associated with stolons or tubers at different developmental stages to screen candidate genes controlling the tuberization process, which laid foundations for enriching the regulatory network of tuberization. These were mentioned in the abstract and the last paragraph of introduction, and supplemented in the revised manuscript.

4. Why did the authors use only the tetraploid cultivar ‘Atlantic’ as the experimental material in their study? The gene expression analyses would have a greater strength if the authors have used at least two cultivars in order to validate and confirm the results of their research.

Response: 'Atlantic' is widely planted in regions such as the United States, Canada, Europe, and Asia. It has the advantages of high yield, shallow sprout eyes, and storage resistance, making it a high-quality potato cultivar. Therefore, we believe that using 'Atlantic' as the experimental material for transcriptome research has great reference value. Additionally, we would greatly appreciate your proposal to validate gene expression data using two different cultivars. Unfortunately, due to limitations in time, materials, and funding, we have only used one cultivar for confirmation at present.

5. I find that the presentation of results is not well organized. There is a lot of data analysis and information that appear to have no relevance.

Response: In the results section, our presentation logic is as follows: Firstly, analyze the quality of sequencing data to determine its reliability. Subsequently, WGCNA analysis was performed on the expressed genes of all tissues to obtain co-expressed modules associated with stolons or tubers. Then, analyze the differentially expressed genes, transcription factors, and hormone related genes during tuber development. Finally, among the three co-expression modules obtained, genes that were not differentially expressed during tuber development were removed, and a co-expression network associated with tuber formation was constructed. Finally, hub genes and hub transcription factors in the networks were screened as candidate genes for regulating different stages of tuber formation.

Reviewer 3 Report

Comments and Suggestions for Authors

The manuscript presents a valuable investigation into the transcriptomic regulation of tuberization in potato plants. However, it requires significant revisions to improve clarity, specificity, and depth of analysis. I recommend a major revision of the manuscript to address the aforementioned comments and enhance the overall quality and impact of the study.

The authors are encouraged to refine the introduction by providing a more comprehensive review of relevant literature and clearly articulating the novelty and research gaps addressed by the study. Additionally, they should present the results with greater clarity and interpretation, and offer more insightful discussions and conclusions regarding the implications of the findings. These revisions will strengthen the manuscript and better position it for publication in a peer-reviewed journal.

Specific comments:

- The abstract provides a concise overview of the study, highlighting the significance of understanding tuberization in potato plants. It effectively summarizes the methods, results, and conclusions of the research. However, the novelty and specific contributions of the study could be elaborated further.

- The introduction sets the stage for the importance of understanding tuberization in potatoes but lacks a thorough review of existing literature and a clear identification of the research gaps. A more comprehensive discussion on the current state of knowledge regarding tuberization regulation and the specific gaps that this study aims to address would enhance the introduction's effectiveness. Moreover, most of the presented literature is old which do not describe the current state of knolwedge, this literatuer may be upgraded

- While the study contributes to our understanding of the regulatory network underlying tuberization, the manuscript would benefit from a clearer delineation of its novelty. Highlighting how this research builds upon and extends previous findings in the field would strengthen its novelty.

- The discussion section provides valuable insights into the implications of the study findings for understanding tuberization regulation. However, it would benefit from a more in-depth interpretation of the results, including potential mechanisms underlying the co-expression networks identified and their relevance to known regulatory pathways of tuberization.

- The conclusion effectively summarizes the main findings of the study but could be strengthened by offering insights into the broader implications of the research and potential avenues for future investigation

Comments on the Quality of English Language

only minor imporvements needed 

Author Response

We appreciate your critical review and positive appraisal of our manuscript entitled “Transcriptomic network analysis provides insights into the regulation of tuberization” (Manuscript ID: plants-2837595). We have revised our manuscript in accordance with the constructive criticism and suggestions received. The specific reply to your comments are as follows:

1. The abstract provides a concise overview of the study, highlighting the significance of understanding tuberization in potato plants. It effectively summarizes the methods, results, and conclusions of the research. However, the novelty and specific contributions of the study could be elaborated further.

Response: Thank you for your affirmation and valuable advice. We have supplemented and improved the novelty and specific contributions of the research in the abstract section. Compared to previous transcriptome studies, the tissues we sampled are more diverse and the analysis is more in-depth. We not only analyzed differential genes, but also analyzed differentially expressed transcription factors and hormone related genes. The most important thing is that we used WGCNA to construct a co-expression networks associated with different stages of tuber development. Based on this, we screened a large number of candidate genes that regulate tuberization.

2. The introduction sets the stage for the importance of understanding tuberization in potatoes but lacks a thorough review of existing literature and a clear identification of the research gaps. A more comprehensive discussion on the current state of knowledge regarding tuberization regulation and the specific gaps that this study aims to address would enhance the introduction's effectiveness. Moreover, most of the presented literature is old which do not describe the current state of knolwedge, this literatuer may be upgraded

Response: Based on your suggestion, we have added the current research status of the tuberization regulation network and the problems that this study aims to solve in the introduction. In addition, we have updated the references. The updated references are as follows:

[2] Zierer,W.; Rüscher, D.; Sonnewald, U., Sonnewald, Tuber and tuberous root development. Annu. Rev. Plant. Biol. 2021, 72, 551–580. doi: 10.1146/annurev-arplant-080720-084456.

[9] Muñiz,García, M.N.; Stritzler, M.; Capiati, D. Heterologous expression of Arabidopsis ABF4 gene in potato enhances tuberization through ABA-GA crosstalk regulation. Planta. 2014, 239, 615–631. doi: 10.1007/s00425-013-2001-2.

[10] Sarkar, The signal transduction pathways controlling in planta tuberization in potato: an emerging synthesis. Plant CellRep. 2008, 27, 1–8. doi: 10.1007/s00299-007-0457-x.

[29] Hannapel,D.; Sharma, P.; Lin, T.; Banerjee, A. The multiple signals that control tuber formation. Plant Physiol. 2017, 174, 845–856. doi: 10.1104/pp.17.00272.

3. While the study contributes to our understanding of the regulatory network underlying tuberization, the manuscript would benefit from a clearer delineation of its novelty. Highlighting how this research builds upon and extends previous findings in the field would strengthen its novelty.

Response: Thank you for this valuable suggestion. We compared our research findings with previous research in the results and discussion section, and described our extension and development. By comparison, previous transcriptome analysis has shown that DEGs are enriched in energy metabolism, carbohydrate metabolism, hormone signal transduction, and flower development during tuber development [23,24], which is consistent with our findings. In addition, we also found that a large number of genes that were specifically expressed or differentially expressed in stolons and tubers were enriched in cell division and differentiation, branching system development, cellulose metabolic process, radial pattern formation and nitrogen utilization process.

4. The discussion section provides valuable insights into the implications of the study findings for understanding tuberization regulation. However, it would benefit from a more in-depth interpretation of the results, including potential mechanisms underlying the co-expression networks identified and their relevance to known regulatory pathways of tuberization.

Response: Thank you for this valuable suggestion. We have made modifications and improvements to the discussion, and summarized the potential mechanisms of each co-expression network. Through analysis and discussion, we found that there were significant differences in the potential mechanisms of the three co-expression networks. Genes in MEdarkgrey (Sto–Sto1) participated in tuberization by controlling sink-source balance, cell division, hormones, and flowering signals. The genes in MElightsteelblue (Sto2–Tu1) were mainly involved in hormone synthesis, metabolism, and signal transduction. However, the genes in MEantiquewhite2 (Tu2–Tu5) regulated cell division and tissue differentiation, energy storage and metabolism, which affect tubers maturation and enlargement.

5. The conclusion effectively summarizes the main findings of the study but could be strengthened by offering insights into the broader implications of the research and potential avenues for future investigation

Response: Thank you for your affirmation and valuable advice. We have added potential regulatory pathways for future research in the conclusion section of the revised manuscript. The conclusion has modified as follows, with the yellow highlighted parts indicating the modified content:

In this study, transcriptome analysis was performed in 13 tissues of potato plants at the tuber proliferation phase. Among them, eight tissues were stolons and tubers at different developmental stages, which were focused. Numerous DEGs were screened at different developmental stages of stolons and tubers, which contained many TFs and hormone pathway genes. Simultaneously, three co-expression networks associated with stolons and tubers were constructed using WGCNA. Through analysis and discussion, we suggested that the HGs, HTFs, and co-expressed genes in these networks may regulate processes such as sucrose transport, hormone synthesis, metabolism, and signal transduction, cell division and tissue differentiation, sink-source balance, energy storage and metabolism, and stolon branching, or serve as tuberization signaling to control tuber formation and development. The research results provide valuable references for further revealing and enriching of regulation network of tuberization.

Round 2

Reviewer 2 Report

Comments and Suggestions for Authors

The authors have made improvements on the manuscript, but English language and syntax is not always clear, while the presentation of results and discussion needs revision.

In addition, as I have mentioned in my previous review, these data have not been validated in a second cultivar, therefore these are only preliminary data that have not been confirmed in another genetic background.

Comments on the Quality of English Language

moderate editing of English language is required

Author Response

Thank you very much for your re-review of our manuscript. Respond to your feedback as follows:

The authors have made improvements on the manuscript, but English language and syntax is not

always clear, while the presentation of results and discussion needs revision.

Response: We have revised and polished the grammar and English language of the entire manuscript. At the same time, the results and discussion sections were revised.

In addition, as I have mentioned in my previous review, these data have not been validated in a

second cultivar, therefore these are only preliminary data that have not been confirmed in another

genetic background.

Response: Due to time constraints, we did not perform data validation in the second cultivar by qRT-PCR. However, we compared the expression of some genes from the three co-expression networks in the RH89-039-16 transcriptome (http://spuddb.uga.edu/pgsc_download.shtml) and the Atlantic transcriptome (Supplementary Figure S1). The expression patterns of most genes were highly similar in the two transcriptomes, further confirming the reliability of our result.

Reviewer 3 Report

Comments and Suggestions for Authors

Manuscripts is well improved 

Comments on the Quality of English Language

Fine 

Author Response

Thank you very much for your re-review of our manuscript.